# From Land to Sea, a Review of Hypertemporal Remote Sensing Advances to Support Ocean Surface Science

**Rory Gordon Scarrott** [1,*], **Fiona Cawkwell** [2], **Mark Jessopp** [3], **Eleanor O'Rourke** [4], **Caroline Cusack** [4] **and Kees de Bie** [5]

1  Department of Geography, and Environmental Research Institute, University College Cork, T12K8AF Cork, Ireland
2  Department of Geography, University College Cork, T12K8AF Cork, Ireland; f.cawkwell@ucc.ie
3  School of Biological, Earth & Environmental Sciences, and MaREI Centre Environmental Research Institute, University College Cork, T12K8AF Cork, Ireland; m.jessopp@ucc.ie
4  Oceanographic Services, OSIS, Marine Institute, Rinville, H91 R673 Oranmore, Ireland; eleanor.orourke@gmail.com (E.O.); caroline.cusack@marine.ie (C.C.)
5  Department of Natural Resources, Faculty of Geo-Information Science and Earth Observation, University of Twente, 7514 AE Enschede, The Netherlands; c.a.j.m.debie@utwente.nl
*  Correspondence: r.scarrott@ucc.ie

**Abstract:** Increases in the temporal frequency of satellite-derived imagery mean a greater diversity of ocean surface features can be studied, modelled, and understood. The ongoing temporal data "explosion" is a valuable resource, having prompted the development of adapted and new methodologies to extract information from hypertemporal datasets. Current suitable methodologies for use in hypertemporal ocean surface studies include using pixel-centred measurement analyses (PMA), classification analyses (CLS), and principal components analyses (PCA). These require limited prior knowledge of the system being measured. Time-series analyses (TSA) are also promising, though they require more expert knowledge which may be unavailable. Full use of this resource by ocean and fisheries researchers is restrained by limitations in knowledge on the regional to sub-regional spatiotemporal characteristics of the ocean surface. To lay the foundations for more expert, knowledge-driven research, temporal signatures and temporal baselines need to be identified and quantified in large datasets. There is an opportunity for data-driven hypertemporal methodologies. This review examines nearly 25 years of advances in exploratory hypertemporal research, and how methodologies developed for terrestrial research should be adapted when tasked towards ocean applications. It highlights research gaps which impede methodology transfer, and suggests achievable research areas to be addressed as short-term priorities.

**Keywords:** hypertemporal; Earth Observation data; remote sensing; methodologies; oceanography

## 1. Introduction

Single-date and multi-date remote sensing imagery are widely used in support of oceanographic and fisheries research and monitoring. With increases in the temporal frequency of imagery, a greater diversity of ocean surface features which shape fisheries can be studied, modelled, and understood in terms of their development over time. The advent of the hypertemporal image resource allows researchers to analyse vastly more information-rich datasets (Figure 1). These contain very detailed information on spatially extensive and temporally variable areas, enabling studies to capture the full lifecycle of fast-moving ocean surface features such as eddies, current deviations, and the impacts of wind-driven mixing in surface waters. In 1995, Piwowar and LeDrew [1] noted that the remote

sensing community was on the brink of a "temporal data explosion", as the archive length for some global remote sensing data rapidly approached the 30-year mark deemed necessary to establish climate norms. Over twenty years later, the era of Big Data has arrived [2]. We are currently in the midst of the envisaged explosion. Global datasets at near-daily resolutions are now established (Table 1) and the range of measured parameters is diversifying, which could be invaluable for monitoring the structure and functioning of Earth systems. These are at spatial and temporal resolutions which would have been inconceivable 25 years ago, with climatologically relevant temporal extents. While a large body of work exists on the terrestrial application of these temporally long and dense time series of remotely sensed datasets, ocean applications have received less attention.

**Table 1.** A sample of existing hypertemporal datasets for ocean applications.

| Product | Parameter | Temporal Extent | Temporal Resolution | Spatial Resolution | Described In |
|---|---|---|---|---|---|
| GHRSST Global Ocean Sea Surface Temperature Multi Product Ensemble (GMPE) | Sea surface temperature | 2009–present | Daily | ~0.25° | Martin et al. [3] |
| MODIS Aqua Chlorophyll-a Concentration Level 3 | Sea surface photosynthetic activity | 2002–present | Daily | ~4 km$^2$ | NASA [4] |
| Sea Ice Concentrations from Nimbus-7 SMMR and DMSP SSM/I-SSMIS Passive Microwave Data, Version 1 | Sea ice | 1978–2018 | daily | ~25 km$^2$ | Cavalieri et al. (updated yearly) [5] |
| Global Wind Level-3 ASCAT 12.5 km Coastal Wind Product | Surface winds | 2012–present | daily | ~12.5 km$^2$ | Vogelzang & Stoffelen [6] |

Hypertemporal data are essentially multitemporal data, collected with a very fine temporal resolution (Figure 2). Piwowar and LeDrew [1] were amongst the first published authors to use the term hypertemporal (also referred to as "hyper-temporal", or "high temporal resolution") with respect to satellite remote sensing data (hereafter referred to as Earth Observation, or EO, data). They also highlighted the need for new hypertemporal image analysis approaches to process temporal signals in these datasets in a spatially coherent manner. Such approaches could include novel techniques and algorithms, but also draw upon the pool of existing methods, adapted for use on large, temporally orientated datasets (see Figure 2). Despite their efforts and the increasingly frequent reference to hypertemporal data in the scientific literature, the term is still somewhat ill-defined. Jakubaukas et al. [7] noted a dataset which may require hypertemporal analysis can contain upwards of several dozen to several hundred successive digital images, echoing Piwowar and LeDrew's [1] sentiments. Drawing on an analogy of hyperspectral data, it is essential to consider the characteristics of what makes such data hyperspectral as opposed to multispectral. Hyperspectral digital images are spectrally overdetermined with more data points than are realistically needed [8] for an analysis. Whilst they feature a high degree of data redundancy, and potential information duplication, this characteristic provides ample spectral information to identify and distinguish between spectrally similar, but unique materials. Echoing the interpretation of hyperspectral, Kleynhans [9] provides useful guidance on the term hypertemporal, interpreting it to mean "highly sampled", taken at regular, constant intervals, usually a few (8–30) days apart. They specifically highlight the importance of "frequent, equal spaced observations".

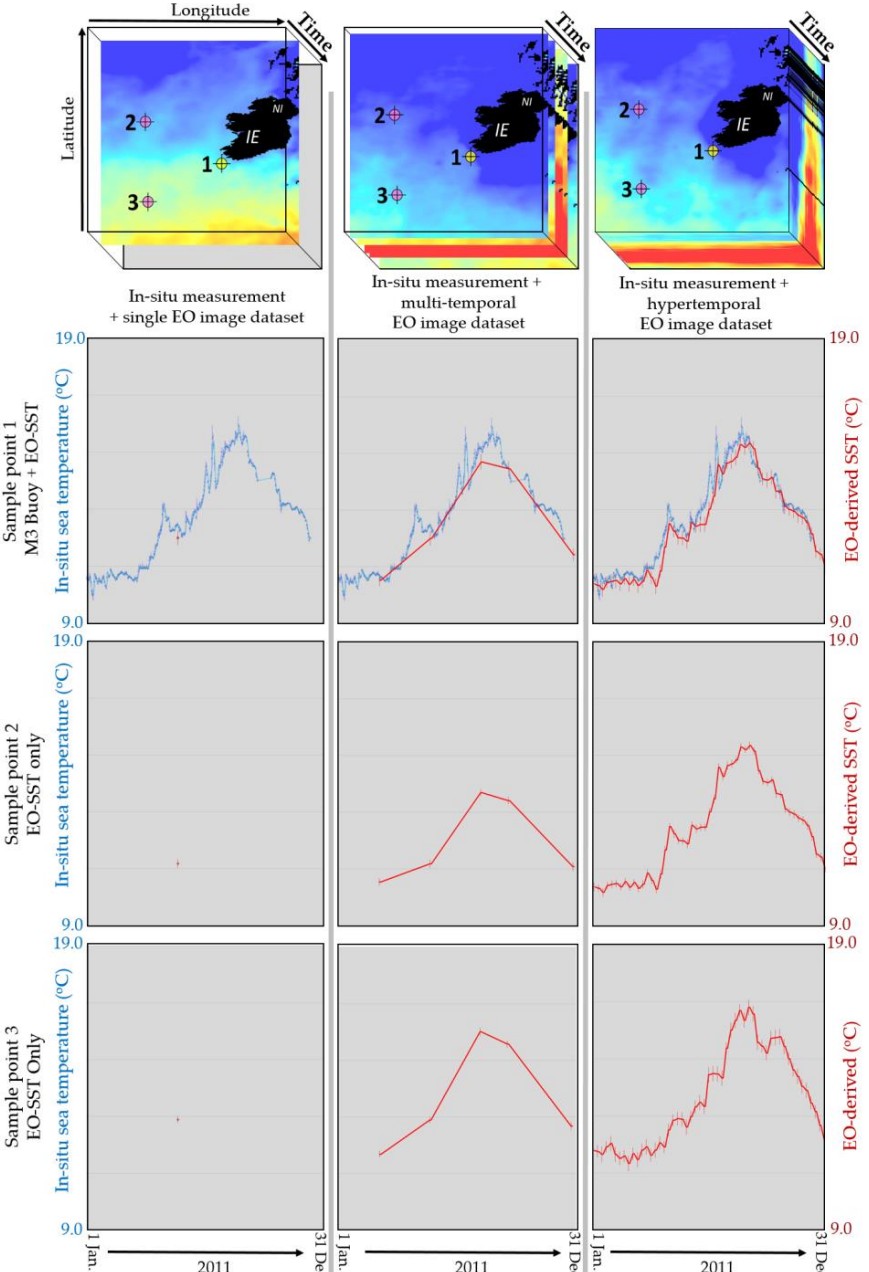

**Figure 1.** Hypertemporal data versus those of reduced spatial extent (in-situ data only) and temporal resolution. Shown are available in situ sea temperature data for 2011 versus Earth Observation (EO)-derived estimates of sea surface temperature (SST) for three sampling points near Ireland. Single-image, multi-temporal, and hypertemporal cases are arranged as three columns consisting of an image visual, and the data available for each sample point. Sampling point 1 can avail of both EO-derived (from the level-4 GHRSST product) and in-situ data (obtained from temperature sensors on board the M3 weather buoy). Sample points 2 and 3 can avail of EO-derived data only.

It is the temporal density (number and frequency) of repeated measurements, as well as the character of measurements in relation to each other, which qualify data as hypertemporal. While sensors can measure hundreds of bands, it is the narrowness and contiguous nature of the band measurements that make them hyperspectral [8]. For hypertemporal images to be useful, the data need to be consistent from time-slice image to image [10], just as internal image consistency is often assumed in the analysis of multispectral images [11]. Regarding hypertemporal data, each estimate of

a parameter is essentially a point measurement in time, acquired over a maximum of a second, with the contiguous nature established by the revisit period of the satellite platform. This review only considers satellite-derived hypertemporal data conforming to the stipulations outlined by Piwowar et al. [12]. Specifically, the input data analysed in a study must:

1. Be univariate in nature (with multiple images of the same parameter only);
2. Contain a set of time slices, all of which must be precisely co-registered (with image-to-image pixels perfectly aligned spatially); and
3. Exhibit radiometric consistency between images (i.e., they are measured using the same sensors or inter-validated sensor systems, and exhibit a degree of normalisation between time slices).

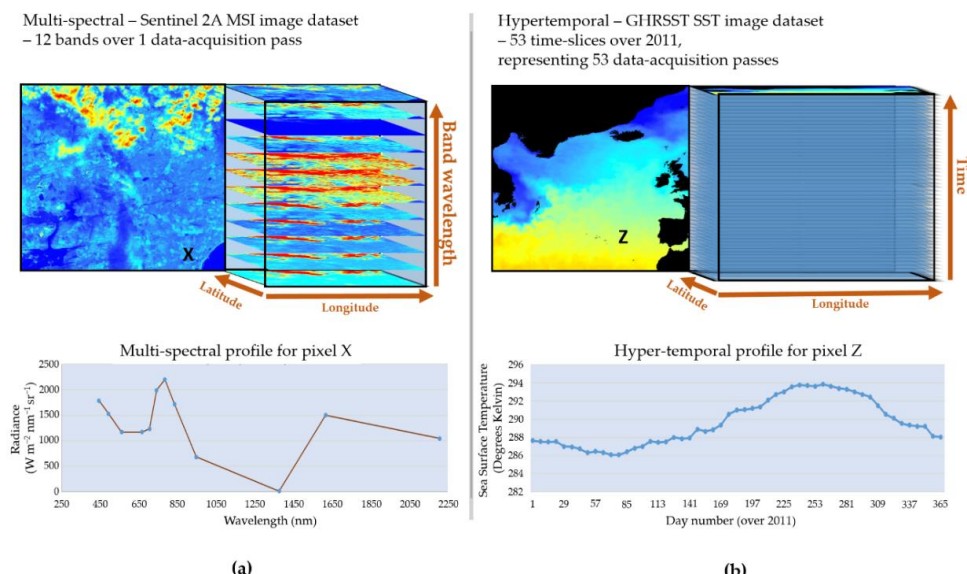

**Figure 2.** Moving from a spectral to a temporal logic to exploit hypertemporal imagery using existing methodologies. (**a**) Multispectral image data from a single Sentinel-2A MSI acquisition over Southern Ireland on 10 February 2019, with the multispectral signature at point X displayed below. (**b**) Hypertemporal data from 1 year (2011) of level-4 GHRSST sea surface temperature data, processed for the North Atlantic Region, with the hypertemporal signature at point Z displayed below. Both three-dimensional datacubes (latitude, longitude, and either band or time) can contain pixel-based signatures, with the temporal signatures containing the inherent sequencing provided by time. Methodologies founded upon a spectral logic can therefore be adapted along a temporal logic.

Since 1995, a diverse range of applications have been developed which involved extracting meaningful information from hypertemporal data. These include landcover mapping [10,13], landuse mapping [14], change detection [15–18], ecosystem structure and species modelling [19–23], phenology mapping [24–26], gradient analysis [27–29], and data quality assessment [30]. To date, a limited range of ocean-focused hypertemporal studies have been conducted (Figure 3), hampered by validation challenges and limited knowledge on the partitioning of temporal patterns over the ocean's surface. Hypertemporal altimeter data have been used to study eddy propagation routes and velocities [31]. Work on sea ice has exploited hypertemporal microwave-derived sea ice concentration data acquired over Arctic seas [32]. A range of different approaches have been trialled, overcoming both technical and theoretical challenges whilst doing so [12,31–33]. It is recognised that the research community are not fully exploiting the information content of satellite observations [34]. New theoretical frameworks are required to better exploit high-resolution information from satellite data. This review explores the limited oceanic work which has been done, and highlights the potential to build upon the more extensive terrestrial experience available. It suggests priority areas of research to catalyse the use of hypertemporal datasets of satellite-derived parameters in future oceanic work.

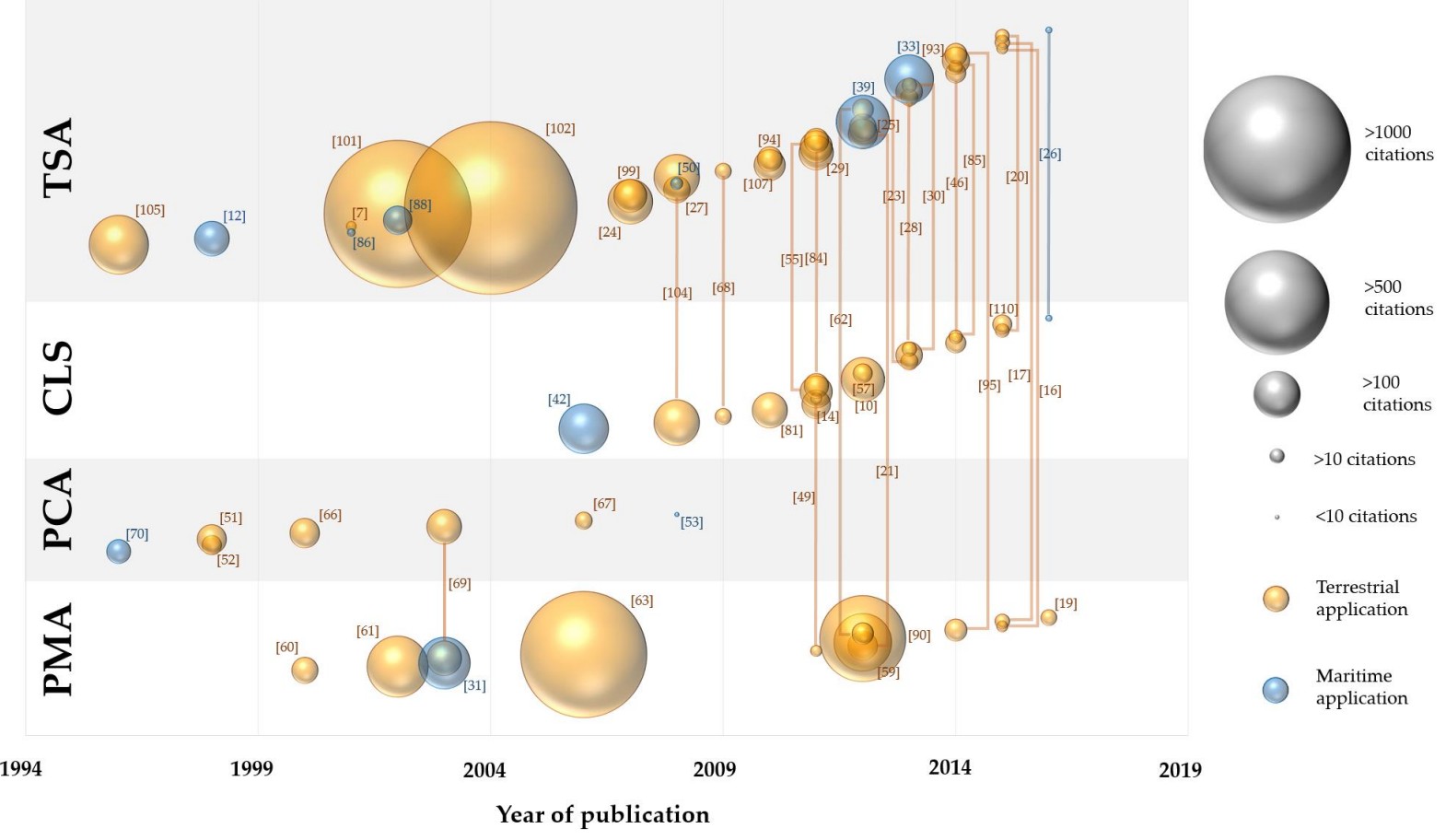

**Figure 3.** Methodological nature and grey-literature impact of hypertemporal papers reviewed in this article. Peer-reviewed articles and book chapters which implemented a methodology on a hypertemporal dataset (as defined here), have been classified on the basis of the outlined four primary approaches, and whether they were focused on a marine or terrestrial application. The bubble area infers the grey-literature impact, which was determined by recording the number of Google scholar citations associated with each publication on the 2 September 2019. Studies which deployed composite methodologies are indicated by linking lines between the component methodologies. Bracketed numbers link to the reference list of this review, to aid the reader identify the specific publication of interest to them.

## 2. Challenges and Opportunities for Hypertemporal Remote Sensing

The Earth's ocean, covering approximately 71% of the planet's surface [35], is a challenging and expensive area from which to secure in situ measurements at a high spatial and/or temporal resolution. Detailed information on the spatiotemporal characteristics of the ocean surface, therefore, are rare in comparison to those found on the terrestrial realm. One can simply compare the range of data sites and species which are available in terrestrial phenological databases such as the USA National Phenology Network database [36], the Pan-European Phenology database [37], or the Chinese Phenological Observation Network [38], with the lack of an oceanic equivalent, to understand the scale of the observation issue. In the absence of ground-based networks of in situ data, validation of satellite image products remains a significant challenge that marine spatiotemporal studies have yet to adequately address.

Nevertheless, the need for a better understanding of the spatial and temporal variability of ocean surface characteristics prevails. Certain locations at the ocean surface do exhibit prominent changes in biological productivity on a seasonal basis (e.g., [33,39,40]). Others can exhibit large inter-annual variations (e.g., [40,41]), with seasonalities often noted as being multi-modal (e.g., [41,42]). These fluctuations can vary spatially and temporally between years and over seasons. For example, dust transport and deposition can drive high spatiotemporal variability in ocean productivity and surface processes [43,44]. The situation is further complicated by the mobility of the ocean's surface, where mesoscale features such as eddies can move over 4 km per day [31]. With respect to hypertemporal datasets, this high mobility can be recorded across a number of pixels' seasonal signals, and can be expressed within and between different parameters. This presents opportunities to examine, and use, the characteristic internal variability of hypertemporal datasets. For example, variables that characterise the heterogeneity of a dataset can facilitate applications in support of ocean surface partitioning [45], defining habitat boundaries as done for terrestrial vegetation by Ali et al. [46]. They could also complement existing studies which compare species distributions to satellite-derived front locations (e.g., [47,48]).

The long-term satellite image archives now approaching, and in some cases exceeding, 30 years, also provide opportunities. Environmental normals and trends in ocean surface processes, related to hemispheric teleconnections and climatic changes, can be identified and quantified (e.g., [49,50]). Given their univariate and continuous nature, hypertemporal datasets are particularly well suited to providing a statistically robust record. They also represent an opportunity to explore the impacts of fine-scale temporal and spatial fluctuations in surface waters and their oceanographic and biogeographical implications, at various spatial and temporal scales, and timeframes. Maximising the potential of ocean hypertemporal remote sensing requires either (i) an extensive and comprehensive collection of in-situ data and knowledge, or (ii) the adoption of data-driven approaches to generate a library of such signals. When combined with existing in-situ datasets, temporal signal libraries would provide an important resource to support future work. The techniques discussed within this paper are not an exhaustive list, but do represent some suggested approaches to addressing these and other application opportunities using hypertemporal satellite imagery.

## 3. Avenues to Extract Information from Hypertemporal Earth Observation Datasets

There has been over 20 years of progress in developing and adapting methodologies for hypertemporal data applications. Hypertemporal researchers often use composite methodologies (see Table 2 and Figure 3). Well-established methods are combined with novel components to extract meaningful information (new data) from oversampled, noise-rich datasets of the Earth's surface conditions. This makes conducting a methodological review to assist in the development of ocean science applications quite challenging. However, Piwowar and LeDrew [1] suggest three primary avenues of methodology development (Table 2), namely, principal components analysis (PCA), classification (CLS), and time-series analysis (TSA). They note these primary methodologies would be useful for exploring the temporal nature of a hypertemporal dataset [1]. A fourth approach also needs to be added to this primary set, this being the pixel-centred measurement and summary analysis of measurement values (PMA). Whilst very few studies deploy these methodologies in isolation, these four primary approaches should be strongly considered for ocean studies using hypertemporal datasets in the short (5 year) to medium term (10-year). At this stage, the methodologies trialled are exploratory in nature. Each approach noted here has its caveats and drawbacks, as well as its successes. These advances, predominantly in the terrestrial arena, do provide analysts and researchers with a range of options to apply to ocean studies. They are presented here to highlight progress, promote discussion, and encourage others to seek novel methodologies in alternative thematics (such as the meteorological arena). General cautions are highlighted, with the onus being on the reader to determine the more study-specific critiques when considering a methodology to adapt for their particular ocean study.

**Table 2.** A sample of hypertemporal studies showing the range of hierarchically arranged applications, and range of data processing methodologies. The hierarchy is aligned with that proposed by Piwowar & LeDrew [1]. Primary techniques are more likely to be most useful for general explorations of the data's temporal nature. These primary techniques host the four methodology categories discussed in this paper (PMA, PCA, CLS, and TSA). Secondary functions are more helpful in explaining the dataset's nature, whilst tertiary functions are applied pre- or post-processing, and often facilitate interpreting the primary analysis outputs.

| Study (Authors, [Reference]) | Publication Year | Application | Primary | | | | Secondary | | | Tertiary | | | | |
|---|---|---|---|---|---|---|---|---|---|---|---|---|---|---|
| | | | Pixel-Centred Measurement & Analysis (PMA) | Classification (CLS) | Principal Components Analysis (PCA) | Time-Series Analysis (TSA) | Fourier Series Analysis | Temporal Metrics & Phenology Metrics | Temporal Mixture Analysis | Wavelet Analysis (A.) | Post-Classification A. | Autocorrelation A. | Correlation A. | Loadings A. |
| Derksen et al. [51] | 1998 | Links between snow cover & atmospheric circulation | | | ✔ | | | | | | | ✔ | | ✔ |
| Derksen et al. [52] | 1998 | Links between snow cover & atmospheric circulation | | | ✔ | | | | | | | | ✔ | |
| Okkonen et al. [31] | 2003 | Mesoscale eddies | ✔ | | | | | | | | | | | |
| LeDrew [32] | 2005 | Sea ice variability | | | ✔ | | | | | ✔ | | | | ✔ |
| Piwowar & Derksen [53] | 2008 | Sea ice concentration & atmospheric teleconnections | | | ✔ | ✔ | | | | | | | | ✔ |
| Piwowar [50] | 2008 | Sea ice concentration and characterising normals | | | | ✔ | | | ✔ | | | | | |
| Kleynhans et al. [54] | 2010 | Landcover classification & change detection | | ✔ | | ✔ | | | | ✔ | | | | |
| Piwowar [49] | 2011 | Characterising normal for vegetation vigour and anomalies | ✔ | ✔ | | | | | | | | | | |
| Salmon et al. [55] | 2011 | Settlement expansion, landcover change detection | | ✔ | | ✔ | ✔ | | | ✔ | | | | |
| Ali et al. [56] | 2012 | Landscape ecology mapping | | ✔ | | ✔ | | | | | ✔ | | | |
| de Bie et al. [57] | 2012 | Landscape heterogeneity mapping, methodology development | | ✔ | | | | | | | ✔ | | | |
| Grobler et al. [58] | 2012 | Landcover classification & change detection | | ✔ | | | ✔ | | | | | | | |
| O'Connor et al. [25] | 2012 | Land surface phenology | | | | ✔ | | ✔ | | | | | | |
| Pittiglio et al. [21] | 2012 | Inputs for species distribution modelling | ✔ | | | ✔ | | | | | | | | |
| Ali et al. [28] | 2013 | Landcover, gradient mapping | | ✔ | | ✔ | | | | | ✔ | | | |
| Girma et al. [20] | 2015 | Species distributions | | ✔ | | ✔ | ✔ | | | | ✔ | | | |
| Kleynhans et al. [16] | 2015 | Landcover change detection | | | | ✔ | | | | ✔ | | ✔ | | |

### 3.1. Pixel-Centred Measurement and Summary Analysis (PMA)

The simplest approach uses either the raw measurements contained in each pixel or everyday statistical measures (mean and standard deviation) to highlight key features of the time series. This can involve the use of thresholds, or exploit summarising the variability into relatively simple and meaningful values (see Figure 4). For example, Okkonen et al. [31] extracted measurements of sea surface height anomalies and geostrophic velocity from hypertemporal altimeter data. Using these anomaly measures, timing, and latitude location, they were able to identify eddy propagation characteristics, charting the routes along which the various propagation velocities occurred. In some cases, it is appropriate to summarise key periods or intervals of the time series. For example, Piwowar [49] determined a 22-year environmental normal of vegetation vigour from NDVI (normalised difference vegetation index—a satellite-derived measure of terrestrial photosynthetic activity) data. This approach can be extended by doing a trend analysis on statistical summaries of different periods. This was demonstrated by Wessels et al. [59] in detecting land surface degradation over the course of a 13-season period captured in an NDVI dataset.

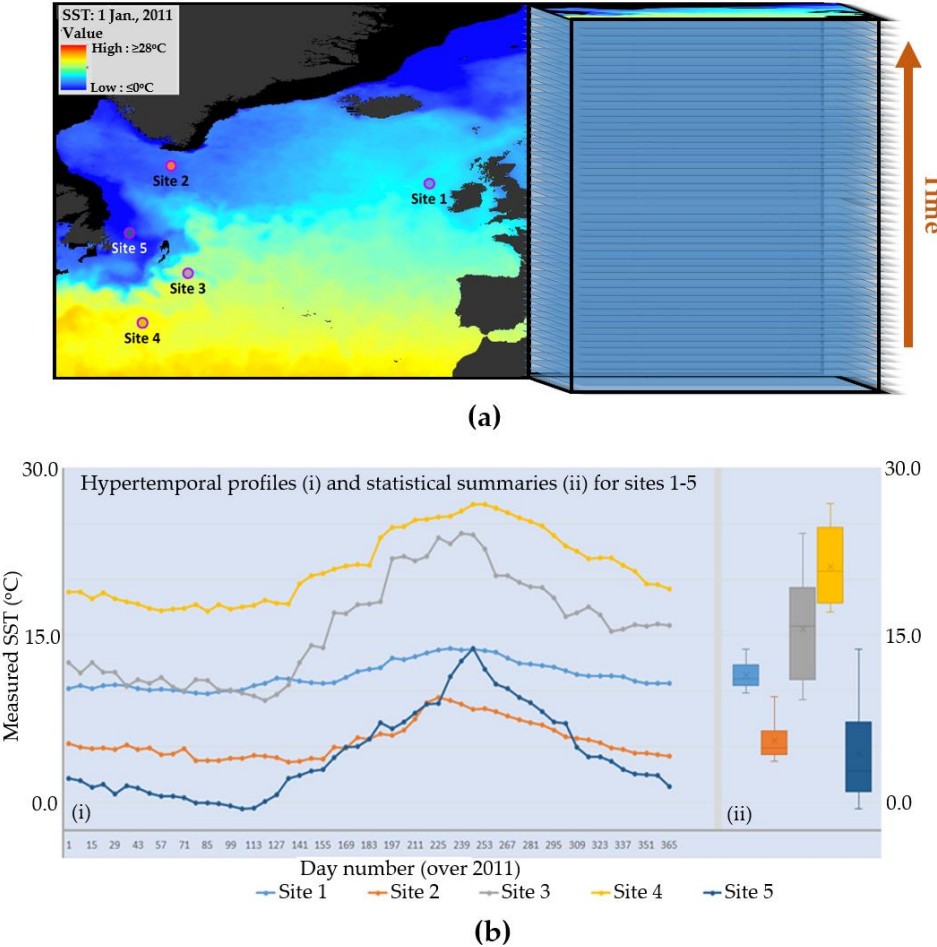

**Figure 4.** Example hypertemporal datacube of GHRSST level-4 sea surface temperature data for the North Atlantic, showing potential distinct signature features and the potential for simple statistical summaries. (**a**) Sample image from the 1 January 2011, showing the dataset extent, and coupled with a 3D representation of the temporal datacube. The locations of sites 1 to 5 from which temporal signatures were extracted are indicated. (**b**) The temporal signatures from example sites 1–5, coupled with box plot statistical summaries of their values.

As output products, the statistical summaries are useful (e.g., [31]), and can be analysed further (as shown by Piwowar [49]). Furthermore, studies in species modelling have demonstrated they can also be combined with data from other sources (e.g., [21,60,61]). However, relative to the potential of higher-order processing, raw hypertemporal measurement values or foundational statistical summaries are of limited use. Studies have demonstrated the utility of exploring more computationally intensive approaches. For example, Kleynhans et al. [62] demonstrated how an autocorrelation-function-based method outperformed an NDVI differencing method [63] when used for change detection. Research has demonstrated the weaknesses of using raw values in examining gradual landcover change, showing that more sensitive analytical procedures were required to cope with weather impacts on the data and trends within the data [59]. Meanwhile, others have shown how statistical summaries of raw values can often incorporate pixel values afflicted by long-term cloud contamination [30]. This is a particular concern when using measurements derived from the optical sensors, and is a driver for the adoption of higher order, noise reducing methodologies. For example, Pittiglio et al. [21] had to use an adapted Savitzky–Golay filter (a form of time series analysis) to model their NDVI time series, producing inputs for a model on elephant seasonal presence. This was an effort to reduce noise from missing values and cloud contamination of their optical-based imagery. Whilst this did introduce a potential subjective influence into the data through the Savitzky–Golay modelling, it did enable them to overcome the cloud contamination issue. This highlights the potential to move beyond raw data, and the use of simple statistical summaries. However, in spite of the limitation associated with using foundational statistical summaries, it is critical to note that they do form the basis of higher-level processing approaches. These seek to extract and use the patterns inherent in the hypertemporal datasets, reduce the data volume, and highlight logical features of the spatiotemporal signals within.

## 3.2. Principal Components Analysis (PCA)-Founded Approaches

Principal components analysis [53,64,65] has proven to be a useful tool to analyse the interrelationships between the highly sampled data within hypertemporal datasets. Typically used as a data compression tool, this data-driven approach undertakes a linear transformation on a set of image bands to create a new band set. This is composed of uncorrelated images, ordered in terms of the amount of dataset variance explained [64], and representing the majority of the information presented in the original time series [53,66]. It can effectively concentrate information from many correlated image datasets into a few uncorrelated principal components [65], highlighting the strongest spatial and temporal signals [53]. The first component captures the characteristic value of that variable within a pixel time series, whilst the second and all remaining standardised components represent change elements of successively decreasing magnitude [64]. Higher-order components thus define more spatially and temporally localised anomalies although they are representative of less "information" from the source data and more noise [53]. The technique has featured in a range of hypertemporal studies focusing on sea ice [1,53], snow cover [66], vegetation studies [67,68], and investigating hypertemporal dataset consistency when derived from multiple sensor sources [69].

In isolation, the components are not very useful, but must be compared and contrasted with other data derivatives to provide meaningful interpretations [54]. For example, Derksen et al. [66] clearly outline how the PCA component eigenvalue ($\lambda$) and eigenvector ($\alpha$) can be used to calculate component loadings for each time slice. A positive loading indicates similarity between the component and the time-slice image, whilst a negative loading indicates dissimilarity. This composite PCA-loadings-plot analysis method highlights not only where important patterns are occurring, but also identifies when they are most prominent, whereby a detailed search for causative factors is done. Eastman and Fulk [64] first used a precursor of the method (which they termed "correlation analysis") on a multitemporal dataset of African vegetation. In less computationally complex approaches, further analysis involves extensive literature surveying for explanatory factors (e.g., [64,70]). However, inputting derived components into further analyses can more rigorously draw out patterns and provide explanations. For example, many studies have examined a spatial pattern's temporal persistence over years through

examining the time-series plot of loadings [66,69,71]. Piwowar and Derksen [53] retrieved the most significant spatial and temporal patterns, then examined the links between the extracted patterns and atmospheric conditions. LeDrew [32] took the outputs from a PCA into a wavelet analysis and subsequent correlations analysis, examining process linkages such as the role of the Arctic Oscillation in forcing of sea ice concentration. Another study by Piwowar [50] took advantage of PCA's capability to minimise inter-band correlation strengths, using the first ten components to inform the development of a refined pixel purity index (adapted from [72]) to aid in temporally un-mixing time series of sea ice data. Meanwhile, Udelhoven et al. [68] demonstrated how PCA could not only produce reduced data for further analysis, but could also receive refined data in the form of time-series profiles smoothed using a Savitzky–Golay filter. They then input selected output components into subsequent decision fusion classifications.

Of all the potential hypertemporal analysis procedures, PCA is particularly well suited to the task of identifying significant anomaly patterns in EO imagery over long time periods [53]. However, the derived principal components are aggregated patterns, and although statistically meaningful, there is no guarantee that the patterns are physically meaningful unless there is extensive visual validation [26]. Loadings analysis can delve deeper and locate a time slice to which the component is statistically similar. This enables the user to refine their search for causative factors to a specific timeframe. However, any such search is therefore founded on the limiting assumption that the spatial patterns in each component are the product of a single factor, isolated from positive or negative feedback interactions between factors. Furthermore, in PCA, a spatiotemporal feature which has a low-magnitude signal, yet is highly prevalent spatially, will dominate the initial components extracted. Signals with a high magnitude, though low spatial prevalence, may feature in later, higher-order components (Figure 5). These may be the signals arising from small but highly important regional features (such as coastal upwellings) or temporary mesoscale features (such as eddies). With PCA, such features may be missed by methodologies which assume higher-order components are of little importance. When the features of interest are mobile over time, such as with mesoscale eddies, this is further complicated. As an eddy's influence moves through the dataset, it has an impact on different pixels at different times, reducing the overall influence on the pixel time series.

Figure 5 shows one such suspected eddy whose lifecycle is clearly identifiable in terms of the surface feature's genesis and conclusion from June to August. When displayed with contoured component images following a PCA of the year-long hypertemporal dataset containing the June to August imagery, the challenges of using PCA to study these features are exposed. In terms of sea surface current data, it is a readily identifiable structure on 2 July, appearing as a ring of surface waters moving at relatively higher speeds. However, this appearance is minimal within the hypertemporal dataset. Despite eddy structures being visible throughout the time-series of imagery, their influence is restricted both spatially and temporally, when and where they occur. This has the effect of relegating their appearance in PCA images to those of higher orders. Furthermore, expressed as sea surface current speeds, the structure's surface expression is composed of both high and low speed areas. When they are accounted for, these variations in speed will be isolated in different components. These are theoretically uncorrelated, demonstrating PCA's inability to coherently capture ocean surface features resulting from a synergy of factors. These spatiotemporal limitations are complicated further by the nature of a PCA (and other hypertemporal methodologies such as CLS and PMA) with regard to removing the order, provided by the unidirectional nature of time, from a hypertemporal dataset. In removing any sequencing in a time-series' temporal progression, PCA's ability to exploit this is effectively curtailed [73]. This renders PCA as being an incomplete method in terms of fully exploiting the spatiotemporal potential of a hypertemporal image dataset, requiring its use as part of a composite process. Components cannot be used in isolation, but are combined with complementary analyses (such as a loadings plot analysis) of the data to derive more complete information extraction. Meanwhile, its utility is severely limited for studying mesoscale features such as eddies without significant adaptation and consideration of the input dataset specifications. Features of interest should first be identified using

other means, before narrowing the spatial and temporal scope of the dataset to ensure the influence of the feature is magnified for the PCA to enhance.

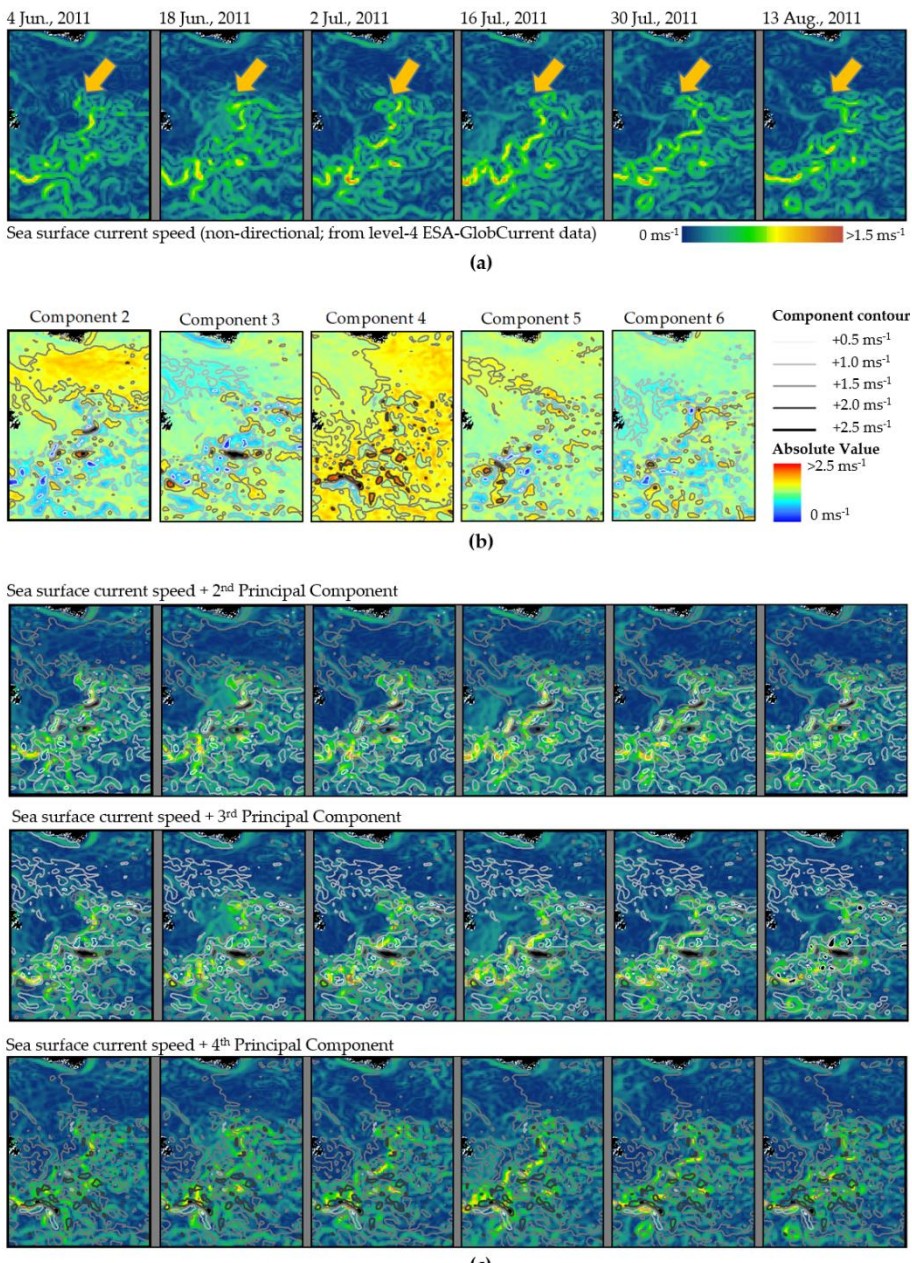

**Figure 5.** Challenges faced by principal components analysis in highlighting mobile mesoscale ocean features in the Grand Banks region of the North Atlantic. (**a**) The evolution of a suspected eddy structure in a subsample of a year-long 7-day interval hypertemporal dataset of level-4 absolute geostrophic speed. (**b**) The imaged and contoured components 2–6 of a principal components analysis of the year-long hypertemporal dataset. Together with component 1, they represent over 86% of the variability contained within the dataset. (**c**) Contoured components 2–4 overlying the sea surface speed imagery, demonstrating the lack of delineation of the feature highlighted in panel (**a**). The speed data have been resolved from a level-4 dataset of geostrophic velocity data produced by the ESA-GlobCurrent initiative and made available through the Copernicus Marine Core Service.

### 3.3. Classification (CLS)-Founded Approaches

Classification involves assigning discrete units (pixels) to single thematic units [74]. Pixels are classified or grouped on the basis of their multivariable statistical properties, or by segmentation based on both statistics or discernible spatial relationships [65]. Analogous to hyperspectral classification, hypertemporal classification uses the pixel's temporal signature. There is a general hierarchy of classification methods, the broadest division being supervised or unsupervised. Supervised classification generally depends on previous knowledge of a study area, acquired by external sources or field work [65,74]. Groups are defined using pixel summary statistics or identified characteristics of training areas representing different objects on the Earth's surface, selected subjectively by users on the basis of their own a-priori knowledge or experience [65]. Unsupervised classification (clustering) approaches identify natural groupings or clusters entirely on the basis of the whole dataset distribution's statistics. This produces an image of statistical clusters for later characterisation a posteriori, by examining their contained area and applying the users knowledge and experience [65,75].

For terrestrial-focused studies, it is possible to collect information on the region's temporal characteristics. Sufficient in-situ data, or expert knowledge can be acquired or collected to drive supervised approaches. In comparison, in-situ data or expertise providing a comparable level of knowledge of the ocean surface's temporal characteristics are few and far between. Furthermore, collection of such datasets can only be achieved at a comparatively high cost, and high risk. This limits the spatial and temporal coverage of data needed to maximise hypertemporal data exploitation. Hypertemporal datasets are also very data rich [57]. Due to the number of bands in the feature space, visual pattern recognition of training groupings is hardly feasible, further limiting our ability to undertake supervised classification studies. If conducting a supervised classification on a hypertemporal ocean dataset, a researcher must effectively identify enough sample temporal training profiles to adequately capture the diversity of the region of interest. This needs to be done with limited, if any, in situ knowledge of the surface situation and history, in an environment in which it is often prohibitively expensive to collect the required data. Faced with such expensive and often insurmountable challenges in obtaining the data or expert knowledge needed to conduct expert-driven classification approaches, researchers must currently focus, initially, on more unsupervised (clustering) approaches. These data-driven approaches demand no a-priori knowledge of the study area [74,75], and are more appropriate at this stage and time for ocean studies. Focusing on describing ocean regions with their temporal characteristics, and clarifying what they represent, would enable subsequent research to exploit more targeted supervised approaches. Research using unsupervised methodologies, or adapted methodologies, has included the use of ISODATA clustering [13,20,28,30], k-means clustering [49,55,76], minimum error [77], maximum likelihood [78], Ward's method [55], and expectation maximisation [55]. k-means and ISODATA clustering are by far the more commonly used approaches.

Both k-means and ISODATA are iterative procedures, first assigning arbitrary initial cluster vectors, then classifying each pixel to the closest vector, calculating new cluster means, and then iteratively repeating the cycle until the change between iterations is deemed to be small. With k-means, the number of clusters remains static throughout, whilst ISODATA allows for a cluster to be split into two, and two clusters to be merged on a threshold basis [79,80]. Examples of hypertemporal k-means studies are somewhat limited in number. Piwowar [49] used a hypertemporal k-means classification of NDVI data to define and characterise terrestrial phenoregions. Others input time-series state vectors extracted from a hypertemporal dataset into 2-class k-means clustering [76], demonstrating the potential for an entirely unsupervised change detection approach using hypertemporal data. k-means clustering has also been used as a benchmark against more advanced supervised terrestrial change detection algorithms [55].

A more adaptive method of clustering is the ISODATA (iterative self-organising data analysis) technique. First proposed for hypertemporal research by de Bie et al. [13], subsequent years have seen refinements and tests of the algorithm's use and modes of deployment on a number of different landcover [10,13,56,81] and species modelling applications [20,23]. A primary concern regarding

unsupervised classification is the lack of knowledge beforehand of how many clusters best represent and capture the variability in the hypertemporal dataset [57]. The more clusters the dataset is divided into, the better the fit to the data. However, beyond an optimal cluster number, each additional cluster does not explain a meaningful amount of the variability. This is potentially addressed by taking an ensemble approach to ISODATA classification, using separability (divergence) statistics analysis and identification of coincident peaks to guide identification of this number [28,30]. Throughout its development, this divergence-guided ISODATA clustering approach has not always adhered to the coincident peak rule (e.g., [10,56]), remaining subjective at a key juncture—the analysis of the divergence statistics. There are often experience-driven assessments of a significant peak in average divergence denoted cluster number selection. For ocean deployment, the approach would need to be completely objective, indicating that a high number of cluster outputs may be required (e.g., potentially up to 200 clusters as run by Girma et al. [20]), whilst peaks must be identified using an entirely automated approach. Such an advance would effectively enable ocean researchers to circumvent the in-situ data gap and build the needed a-priori knowledge for more supervised efforts to be undertaken. Automation of this algorithm for ocean use could open some interesting avenues for ocean studies. For example, de Bie et al. [57] integrated the divergence-guided ISODATA clustering algorithm into their novel land heterogeneity mapping (LaHMa) algorithm for quantifying landscape heterogeneity—the variation of a landscape property across space and time [22,82]. This was later used as a basis for investigating landcover gradients underpinning the Cretan landscape [46]. They used the accompanying optimal cluster map to determine a stratified sampling strategy for collecting complementary in-situ data. For ocean sciences the benefits would be twofold. Firstly, stratifying sampling regimes offers cost efficiencies by providing data-driven guidance for sampling transects. Secondly, with sufficient characterisation of surface waters at various scales, it could become feasible to deploy more supervised approaches for which example deployments are evidenced in the terrestrial realm. Possible methodologies could include the use of artificial neural networks [83–85] and decision tree classifiers [26] on hypertemporal datasets, or more likely, hypertemporal dataset derivatives. Furthermore, categorisation of the ocean's surface into regions exhibiting different temporal profiles could provide opportunities for these profiles to be examined and explored. Terrestrially, this has been demonstrated for deriving regionalised crop calendar information [10], and to determine heterogeneous and relatively homogenous regions before investigating their sub-regional surface characteristics [46]. Being able to determine regions of relatively homogeneous temporal behaviour could also aid in characterising regional time-series, enabling an expanded range of methodologies which rely on expert knowledge for their use.

However, unsupervised CLS algorithms tend to use statistical summaries of the temporal profiles which remove the unidirectional nature of time to underpin the clustering process. Whilst this benefits studies which focus on locating extreme conditions, similar to PCA, the approaches inform on *what* is occurring in the dataset, but not *when* it occurs (Figure 6). The ordering of temporal data provided by the unidirectional nature of time remains underexploited. With PCA, an analysis of temporal loadings can provide some insights into when certain component features are most expressed. However, the underlying assumption that each component is the product of a single factor weakens the strength of using this. Furthermore, the analyst must be extremely cautious when using multi-annual time-series of data, staying highly aware of the potential impact of surface changes on their data. Terrestrially, such changes may be the product of landcover or even landuse changes. However, with regard to the ocean surface, long-term gradual changes due to climate change, or even short-term subsurface perturbations such as undersea volcanic activities, can change the signals recorded in ocean parameter measurements between years. This can affect the clarity of the classifications obtained and the accuracy of their interpretation. To address the when, one possible avenue involves using sliding windows [55] to analyse sequential subsections or subsection extracts of the temporal profile. These subsets can then form inputs for classification and applications such as change detection [55], building the connection between what and when. With hypertemporal datasets, this would magnify the computing power

required significantly and, thereby, the cost of undertaking such analyses. Alternatively, research can focus on the use of time series analysis to examine the patterns which occur within a hypertemporal dataset and determine when they are expressed.

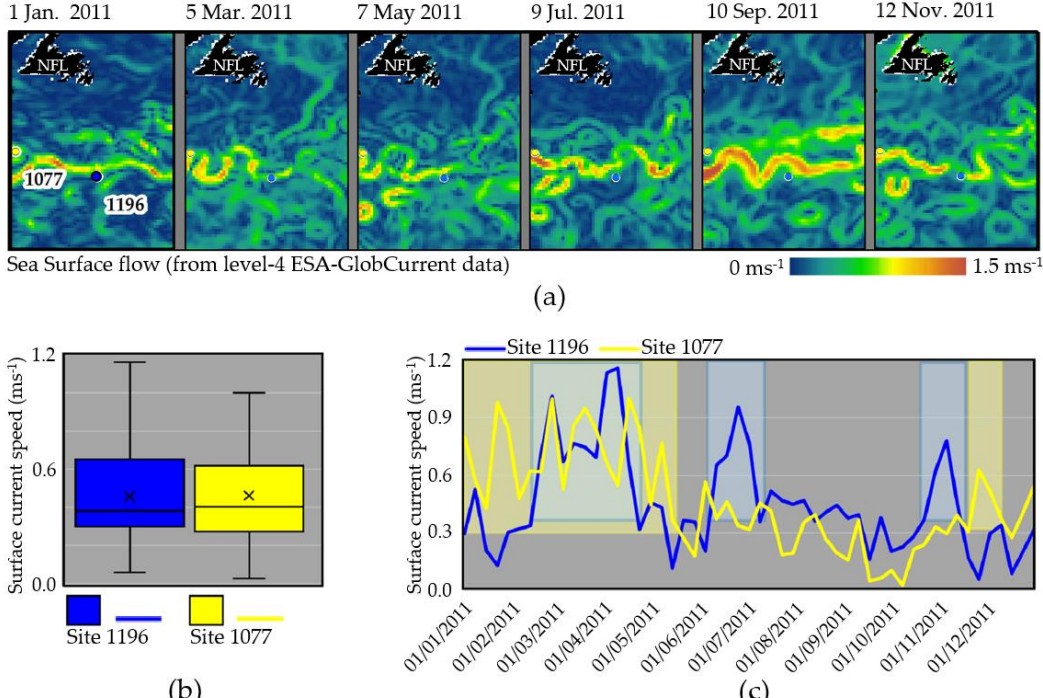

**Figure 6.** Using the unidirectional nature of time to differentiate statistically equal sites. Panel (**a**) shows a sample of surface water flows off the coast of Newfoundland over 2011, extracted from a year-long 7-day interval hypertemporal dataset of level-4 absolute geostrophic velocity data (from the ESA-GlobCurrent initiative and made available through the Copernicus Marine Core Service). Two sample sites (1077 and 1196) are indicated, from which a 53-measurement time series has been extracted covering 2011 at 7-day intervals. Panel (**b**) shows the statistical summary of both sites, which are equal statistically (*t*-test, *t* = −0.083, *p* > 0.2). Panel (**c**) shows how the temporal sequence of peaks and troughs can enable the two sites to be differentiated, in spite of their statistical similarity when the ordering provided by time is removed. The three main peaks for site 1196 are highlighted by light blue panels, whilst the two main peaks for site 1077 are highlighted by yellow panels.

### 3.4. Time Series Analysis (TSA)-Founded Approaches

Time series analysis, the procedure of fitting a model to a given time-series [86], forces a critical evaluation of the temporal data to be examined based on a sound understanding of the phenomena being modelled, an appreciation of the mathematical attributes, and limitations of the models being considered [87]. TSA has three principal objectives [88]: (i) description of the various statistical inputs to the time series, (ii) explanation of the mechanisms which generate the series, and (iii) prediction, which can only be made after (i) and (ii) have been satisfied. It is in the synthesis of the three objectives of TSA that the real power of the process is realised [88]. TSA modelling is a manual process, with the researcher heuristically evaluating the model using graphical displays of summary statistics at each step in the modelling process. The residual series (modelled values—observed) is then analysed by the researcher to determine any remaining autocorrelation/trends not adequately described by the model [86,89]. In this way, the researcher is enabled to learn about the temporal characteristics of their data in a hands-on manner. Studies have demonstrated a range of TSA applications to hypertemporal datasets, including trend analyses [27,90], characterising productivity pulses and drivers [33], mapping phenological variation [24,25,39,91], characterising features of the Earth's surface [29,92], studying

species movement behaviour and migration [93,94], and identifying temporal characterisations for species distribution models [20,95].

With a TSA, the researcher attempts to explain and characterise the occurrence of observed temporal phenomena [96]. When properly constructed, TSA can be useful to identify other temporal sequences of the same process (e.g., [86,89]). Modelling the pixel time-series correctly is an essential part of TSA to which a number of different modelling approaches have been applied, including autoregressive moving average (ARMA) models [88,97,98], Gaussian models [25], adaptive Savitzky–Golay filter [23,28,30,94], Fourier transformation-based approaches [7,90,92,99], and extended Kalman filter approaches [54,100].

Approaches such as ARMA modelling or Savitzky–Golay filtering involve fitting functions to a time-series of data points, and replacing the original values with values predicted by the functions. This increases the signal-to-noise ratio of the temporal data [101,102]. They have been shown to be highly useful when studying landcover seasonality and phenology [25,98], extracting information on sea ice seasonality [88,89] and producing smooth, gap-free data for further processing [28]. Whilst the strengths of these approaches are well documented, their applicability to ocean studies are limited firstly by their weakness in requiring a-priori knowledge to guide model training. For example, the results of an ARMA model deployment on hypertemporal sea ice data directly challenged the assumptions of many previous studies, and in doing so, provided a deeper understanding of the cryospheric processes in the Arctic [86]. However, the study noted that modelling would not be possible without a sound understanding of the processes underpinning the system being modelled. Model refinement depends strongly on the initial time-series being sampled to inform refinement, and the ARMA development process is highly manual. This manual and subjective nature of TSA also features in the Gaussian [25] and Savitzky–Golay [28] approaches. The extraction of metrics (such as the start of season date) is also highly subjective, requiring the selection of arbitrary thresholds [25]. Whilst applicable to ocean studies, they face the same challenges of requiring a pre-existing understanding of system processes, and with caution being taken when sampling initial time-series for defining the function.

Secondly, TSA produces modelled data from which metrics or input images for further analyses are derived. Hypertemporal datasets are univariate. It is rare to find researchers processing raw band data (for exceptions see [16,17]). The datasets processed are usually parameter estimates such as sea ice concentration estimates, or estimates of photosynthetic activity. These are derived from the raw measurements of electromagnetic radiation, which have been processed to correct for potential atmospheric interference on the basis of models. Whilst this does allow researchers to study surface phenomena, the more original data (with more limited model influence) is purer, with less introduced model uncertainties. A critical evaluation of the reduced-noise signal is advisable within the context of model uncertainty propagation and determining the validity of the TSA-modelled outputs.

Some TSA methods do enable the temporal nature of the time-series to be studied in a more data-driven manner. Fast Fourier transformation (FFT) is noted as an effective and computationally efficient algorithm to compute discrete Fourier transforms [103], and has been used when evaluating NDVI time-series data [104]. Fourier analysis has been used on hypertemporal NDVI data [7], highlighting that it can be extremely useful for identifying and isolating modes of data variability. Fourier analyses produce a phase and amplitude dataset for each identified harmonic in the signal, determining the dominant harmonics to carry forward. For example, the dominant harmonics have successfully been put into harmonic analysis of time-series (HANTS) analysis [20,105]. However, it has been noted that Fourier analysis works better on longer time-series [99] as it makes the assumption that the input signals are infinitely long [106]. This is somewhat unrealistic given that processed time-series are of finite length. Furthermore, applying fast Fourier transforms to studies involving terrestrial change detection assumes that the underlying process is stationary [107]. This has implications for any ocean study looking for change, or where the temporal processes featured in the area of interest are in doubt (experiencing a change over the time period in question), or where the processes may be more interannually elastic than is currently understood. An extended Kalman filter (EKF) approach has been proposed to detecting nonstationary events in time-series [108]. Indeed, Kleynhans et al. [108]

reported on the comparison of the EKF-based feature extraction method with the FFT-based feature extraction method, finding optimised change detections using the EKF-based approach. Whereas Fourier could be deployed to explore patterns deemed to be stationary over time, EKF, by contrast, was developed specifically for the purpose of change detection in time.

A further potential avenue involves the use of hidden Markov models (HMM), to examine parameter dynamics. A HMM-based approach has been used to analyse vegetation dynamics at large scales [109], using a hypertemporal dataset of land-surface photosynthetic activity (measured using NDVI). Viovy and Saint [109] described not only the process of applying a HMM to hypertemporal data, but also the complexities involved. The HMM model must be trained, directed by a-priori external knowledge of the surface being examined. In this case, HMM definition required knowledge of (i) where savannah landcover existed, (ii) expert knowledge of what characteristic features of the temporal profile represent (such as seasonal characteristics of savannah vegetation), and (iii) information extracted using a classification approach to extract usable time series for interpreting and training the HMM. For ocean surface studies to exploit HMM-based avenues, knowledge of a study region's temporal diversity is needed to provide the state probability inputs that HMMs require. This need for training remains evident in more recent HMM applications which use hypertemporal datasets, such as a study by [110] looking at the potential for HMMs to be used on hypertemporal MODIS datasets for continuous change detection.

Temporal mixture analysis (TMA), whilst not strictly a TSA method, shares many similarities with the Fourier analysis approach. An adaptation of spectral mixture analysis, this approach has been used to study the temporal characteristics of sea ice concentration [111]. The focus of TMA is (i) the extraction of temporal end-members—pure examples of temporal signals that contribute to some degree to all pixels in a hypertemporal dataset, and (ii) the determination of the magnitude to which those end-members contribute to each pixel's temporal signal. End-members refer to extreme, not average, conditions [12], and can be used to define a baseline of temporal variability, or an environmental normal [50]. However, the selection of end-members can be somewhat subjective and can miss important contributory end-members. Efforts throughout the work by Piwowar et al. [12,50,111] document the challenges faced in obtaining end-members in a more unsupervised manner, before applying a data-driven pixel purity index approach to mask unsuitable end-member pixels [72].

Concerning TSA modelling, there are two principal limitations to consider for ocean studies. The first concerns model specifications and accounting for multi-modal seasonality in the ocean surface data. In agricultural systems, satellite-derived time-series often feature multi-modal seasonality (e.g., [10,99], Figure 7). The same is true of natural ocean surface waters. Ji et al. [33] highlighted the presence of bi-annual pulses in primary production in some Arctic waters. This would be expressed in any time-series of satellite-derived photosynthetic activity measurements, and could also feature in any time series analysis of potential contributory factors (such as sea surface temperature or sea state). Similarly, the bi-modality of the Columbia River plume [42] would influence any seasonality in physical measurements of coastal waters acquired in the nearby vicinity, and could drastically affect the accuracy of the model fit. The second limitation concerns the subjectivity of TSA. There have been welcome advances in TSA methods with regard reducing subjectivity in the TSA process (e.g., progress by Piwowar et al. [12,50,111] in determining the purest possible temporal signals). However, in terms of minimising user-interaction and subjectivity, TSA is not yet at the same level of development as PCA or classification.

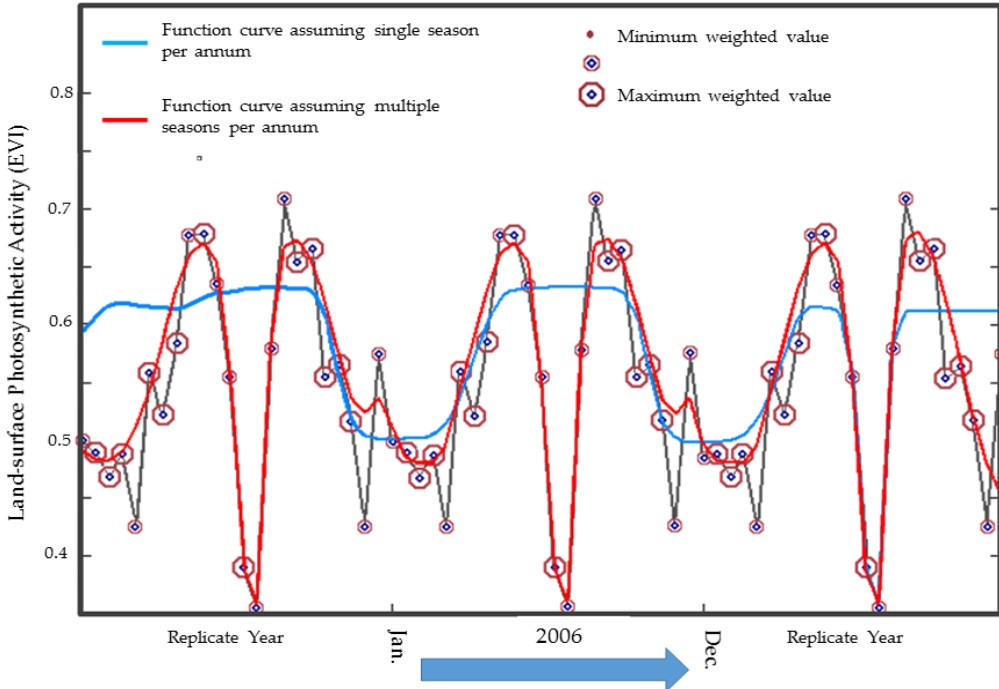

**Figure 7.** Impact of inappropriate model selection on obtaining optimal seasonality profiles for an area of Ireland's photosynthetic activity. Both a single season (blue) and double season (red) Gaussian curve have been fit to a single-year time-series of MODIS enhanced vegetation index data for 2006, with replicates for the years before and after. The presence of a mid-season decrease in photosynthetic activity (potentially linked to intensive field management activities) is obscured by the application of a single-season curve. The conclusions of the modelling shown here are reported in [112].

Where the harmonics extracted by TSA align with an annual cycle, phenological studies of the timing of recurring biological events, the abiotic and biotic drivers of their timing, and the interrelation amongst phases of the same or different species are possible [113]. Justice et al. [114] have illustrated the potential for remote sensing datasets to support investigating vegetation phenology. Subsequent hypertemporal studies demonstrate applications to crop research [10,56], investigations into more natural systems on both land [24] and in the ocean [39], and research into the timing of abiotic events [26]. Satellite-derived phenology metrics characterising the seasonal profile of the hypertemporal time-series, or wider harmonic descriptors (known as "state vectors"), can be used for direct visualisation and analysis [25,93]. They can also be delivered as inputs for further processing [20,26,90,99,115]. The "smoothed time series" itself can also be input into subsequent analyses [28], or characterised further with sequential measures [116,117]. Whilst these higher processing efforts are novel and welcome, the products are modelled data. Researchers must remain conscious that modelling assumptions and subjectivity applied in TSA steps will carry onwards into the post-processing. This is of particular relevance given the potential for the sheer data volume to obscure more subtle influences. Furthermore, for ocean studies interpreting or using derived metrics or harmonics, the user must remain conscious of the validation issues facing temporal ocean studies and the modelling of specification constraints.

In summary, hypertemporal TSA are potentially applicable to any environmental image datasets which meet hypertemporal criteria [88] and can produce highly relevant conclusions to shape understanding of Earth surface processes [86]. When sufficient knowledge of an ocean system is known, the application of TSA can be highly beneficial, as is the case with the range of TSA and TMA studies conducted on Arctic waters. However, TSA does have some limitations which must be considered before use in ocean studies, and which may propagate through further processing chains. The selection of suitable models to fit the time-series data as well as appropriate time-series subsamples to shape these models feature as the strongest caution regarding TSA application to hypertemporal

ocean studies. This is due to a scarcity of in-situ information on the nature of the ocean surface. The range of Arctic studies does highlight the utility of not restricting analyses to a single approach. It demonstrates the potential for conducting multifaceted studies into even a single time-series to discern patterns in the data, and ultimately obtain clarity on the region's spatiotemporal characteristics which are recorded within. An examination of Figure 3 epitomises this, with later TSA studies being composed of multiple methodologies, demonstrating the range of research work achievable by better expert knowledge. In doing so, it hints at the oceanographic knowledge potential which would be unlocked if hypertemporal studies first focused on addressing the knowledge gap on the temporal diversity of the ocean surface.

## 4. Adopting a Strategic Approach for Future Advances

The ability of hypertemporal methodologies to assume a temporal logic makes them quite transferable (with relatively minimal adaptation) to novel time-series datasets (Figure 2). Methodologies developed for land applications may be suitable for use in hypertemporal ocean studies after considering the limitations of each method, and adapting them accordingly. Though advances in both terrestrial and ocean arenas remain very much exploratory, efforts have generated a pool of methods which can be built upon by ocean researchers. Some, such as PCA and CLS, are more readily usable with minimal adaptation. Others, such as TSA, highlight the potential for research once the foundations of expert knowledge are established for the study regions. The approach (methodology blend) taken will depend on the information required. However, for the short- to medium-term, this review highlights three primary considerations to guide studies being undertaken and to ensure best practice in hypertemporal ocean studies. Namely, studies should focus on:

1. Prioritising data-driven approaches;
2. Quantifying temporal signature diversity and ocean surface heterogeneity; and
3. Exploiting the unidirectional nature of time.

These should be integrated into a strategic approach to hypertemporal methodology development and use on ocean surface datasets.

### 4.1. Prioritising Data-Driven Approaches

There is limited information on the mapped temporal diversity of the ocean surface and regional differences. Supervised approaches require not only an expert awareness of the ocean region being captured by the sensor with each acquisition but also an understanding of region-specific spatiotemporal variability. Given the challenges faced collecting in-situ oceanographic data, the application of data-driven approaches should be prioritised in the short (1–5 year) term. This should enable the oceanographic and hypertemporal EO communities to determine what spatiotemporal patterns are evident in the data, or at least indicate the diversity of patterns being expressed in the region. Clarification of this is advisable before proceeding to more expert-knowledge-driven (a priori), supervised and user-interactive approaches.

### 4.2. Quantifying Temporal Signal Diversity and Ocean Surface Heterogeneity

With the limited existing in-situ data available, hypertemporal studies need to be strategic in their planning and execution. A primary barrier to exploiting EO datasets is the dearth of knowledge regarding temporal signal diversity and location. Whilst TSA would appear to be the natural solution to collate these, it can miss unforeseen, yet important, temporal signals expressed over a limited area. It would be more appropriate to first characterise the ocean surface heterogeneity—the variation of ocean surface properties across space and time [22,82]—using variations of CLS and PCA approaches. Qualifying and potentially quantifying the diversity of signals being expressed in the ocean surface measurements will help determine where temporally pure or representative signals may be obtained. Subsequent characterisation of classes, and principal components could enable researchers to more

fully appreciate the range of temporal signals present in their temporal dataset, and refine their TSA approach to integrate this more enhanced, data-driven awareness.

### 4.3. *Exploiting the Unidirectional Nature of Time*

Concerning hypertemporal methodologies in general, the sequential ordering in datasets provided by time represents an underexploited opportunity. Whilst temporal sliding window-based approaches are demonstrating successes with regard to change detection, the temporal sequence is a feature of hypertemporal datasets which defies direct exploitation by CLS- and PCA-based methodologies. Composite methodologies (for example, PCA coupled with loadings analysis) are suggested to feature strongly in advances in this arena, bridging the determination of *what* is occurring and *when* it is occurring by using the ordering provided by time to improve object and pattern recognition. The when is particularly important with ocean research as dispatching resources to sample in-situ measurements of a phenomenon can be extremely expensive with little return if poorly timed.

### 5. Conclusions

A diverse range of approaches now exists to extract useful information from hypertemporal datasets. It is encouraging to note that almost 25 years since Piwowar and LeDrew's [1] call for hypertemporal methodologies, a wide range of approaches has been developed. This gives ocean researchers an extensive pool of knowledge to build upon in studying ocean surface waters. For ocean studies, data-driven approaches should be prioritised until sufficient knowledge of the spatiotemporal patterns of the ocean surface is available to exploit more supervised approaches. This suggests a strategic need to focus on quantifying ocean surface heterogeneity and the diversity of temporal signals being expressed as part of any analysis in the short term. Finally, there is an identifiable gap in hypertemporal research with regards to exploiting the unidirectional nature of time upon which research efforts could be focused.

**Author Contributions:** This research article was compiled with the following contributions from the authors: R.G.S.; Original manuscript preparation, editing and manuscript coordination, F.C.; Supervision, review and editing, M.J.; Supervision, review and editing, E.O.; Supervision, review and editing, C.C.; Supervision, review and editing, K.d.B.; Review and editing.

**Funding:** This research has been funded by the European Union, through the Horizon 2020 research and innovation programme, under grant agreement no. 687289 (Co-ReSyF). For more information on the Co-ReSyF project, see www.co-resyf.eu.

**Acknowledgments:** The authors would like to thank the anonymous reviewers. Their efforts in reviewing this considerably long piece of work, and providing the concise and detailed feedback are very much appreciated.

**Conflicts of Interest:** The authors declare no conflict of interest.

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
