# Peer review of "From Land to Sea, a Review of Hypertemporal Remote Sensing Advances to Support Ocean Surface Science"

_water, doi:10.3390/w11112286_

Round 1
Reviewer 1 Report
This rather long paper seems quite aligned to the topic of the related special issue of the Journal to which it is submitted. In this sense, the subject is interesting. The level of discussion seems sometimes more divulgative than strictly technical, which could be considered as a positive, although perhaps not perfectly suited to a scholarly journal. Anyway, formal consistency and technical rigor could be somewhat improved. The treatment of the various literature examples seems more technique-driven than application-driven. This could be interesting, but in the present case the cited techniques are not exactly cutting-edge (PCA, k-means and ISODATA classification, as well as most of the other cited methodologies, are definitely well-known algorithms used for exploratory data analysis in a wide spectrum of applications), so the overall value of the review risks to be not very high. Also, the subdivision of the kind of approaches, in classes such "direct measurements", "classification", and "principal components" appears rather confused and misleading. In fact, reading on to sect. 3.1, it appears that the "direct measurements" are actually meant as computation of lumped quantities such as mean and standard deviation of the image time series - so why not using a more suited nomenclature? In practice, I understand that the parameter which is used to classify the considered techniques is the way in which the temporal dimension is considered: 1) single statistics for the whole time series of each image pixel or region (mean, st. dev., etc.), 2) highlighting of similar temporal trends (as in PCA), 3) other methods such as supervised / unsupervised classification of pixels based on temporal signatures, 4) (which by the way is not mentioned in the abstract) higher-order time series analysis. It has to be said that such differentiation can be made for any kind of application of hypertemporal data, without any reference to the possible peculiarities connected to the field of the oceanography. I think a more useful analysis would require to define the most important open issues in marine sciences and oceanography which could benefit from the availability of hypertemporal data, and then maybe plug the techniques as possible means to investigate such issues. In this sense, some effort in sampling the existing literature about remote sensing applications to ocean sciences (which is definitely huge) would be of more help in my opinion. On the contrary, listing techniques to be transferred from land to sea applications can be problematic, since, as the authors admit, many problems remain peculiar in marine sciences and thus would require consistent modifications. Some additional observations: The definition of hypertemporal data on lines 54-72 should probably be moved at the beginning of the paper, and also, possibly, in the abstract, citing the analogy with hyperspectral imagery. Although it seems to be rather dated, it is not so immediate to understand what is being treated. Line 67: "more data points than are realistically needed": what do you mean? Table 2 has some problems: the subdivision of methodologies does not seem to be consistent. If the methodologies subdivision is the same as that in the abstract, and in sects. 3.1, 3.2, 3.3 (and 3.4?), then I do not understand why PCA (should be in 2°) and TSA (should be in 3°) appear under the 1° methodology. Also, loadings are one of the outputs of PCA, so again the last item in 3° methodology does not seem to belong here. I also note that Figure 3 reports, instead, 4 types of methodologies - differencing between PCA and CLS. Line 419: this definition of TSA seems too restrictive. TSA does not necessarily involve "fitting a stochastic model". Line 425: "TSA modelling is a manual process": please explain. Line 524: what is meant here by "subjectivity of TSA"? Should, e.g., the "fitting of a stochastic model", as is mentioned earlier, be seen as a subjective outcome? The definition of "asymmetry of time" is rather awkward, in my opinion. From the description in sect. 4.3, it refers to the consideration of the precise time series trends (of course considering the temporal ordering of the data), rather than summary measures such as mean, variance, or frequencies. If this is correct, then most of the content of sect. 4.3 is rather confused. For instance, there is nothing preventing consideration of temporal sequences in PCA, since the actual PCs can well be (ordered) time series, with scores which vary spatially. Finally, note that the citation style is not consistent throughout the paper (author-date vs. numerical labels).Author Response
Please see the attachment.
Responses to all three reviewers have been supplied in a single document, as a number of comments are closely aligned, or needed to be responded to together.

Reviewer 2 Report
This is a very nice piece of work.
Author Response
Please see the attachment.
Responses to all three reviewers have been supplied in a single document, as a number of comments are closely aligned, or needed to be responded to together.

Reviewer 3 Report
This paper is a review of methods for evaluating so-called hypertemporal ocean data, which are variables sampled over a long time period at high temporal resolution. The methods fall into four different types, principal components analysis (PCA), time-series analysis (TSA), classification analyses (CLS) and direct extraction and statistical summation (DES). Many of these techniques are common in studying land processes, but less so in the ocean. Honestly, I had a lot of trouble wading through the unfamiliar terminology, though in a sense that is exactly the point here.
The authors have done a lot of reading, and sorted a large number of papers into what type of method has been used. It seems a very useful resource for scientists looking for unique ways of understanding high resolution satellite datasets. So, I would recommend the paper be published largely as is.
My only caveat is puzzlement at why it has been submitted to this particular journal. An examination of the journal's focus statement and many of the recent articles published (https://www.mdpi.com/journal/water), indicates that this article, while not inappropriate or misplaced, is mis-directed. The articles in this journal are mainly about land applications (wetland biomass, rural water supply, agricultural wastewater, etc.). If the authors are wanting to reach people who study the ocean, there is almost no chance that will happen as this is not a journal commonly read in the ocean science community. There are other more appropriate journals, even by the same publisher. All that said, it is the choice of the authors and the editor as to where this is published, not mine.
I have given a couple of random comments below.
Line 55, etc. I'm not clear on why some references are given with numbers and some with author names. Can the editor clarify the editorial policy?
Lines 87-89. This does not make sense. Something is missing.
Line 104. Applications of what?
Author Response

(The authors gave the same response as above.)

Round 2
Reviewer 1 Report
I thank the authors for the good and large amount of work they put into improving the paper. I believe it is now ready for publication.
This manuscript is a resubmission of an earlier submission. The following is a list of the peer review reports and author responses from that submission.